# High Sensitivity and Specificity of Dormitory-Level Wastewater Surveillance for COVID-19 during Fall Semester 2020 at Syracuse University, New York

**DOI:** 10.3390/ijerph19084851

**Published:** 2022-04-16

**Authors:** Alex Godinez, Dustin Hill, Bryan Dandaraw, Hyatt Green, Pruthvi Kilaru, Frank Middleton, Sythong Run, Brittany L. Kmush, David A. Larsen

**Affiliations:** 1David B. Falk College of Sports and Human Dynamics, Syracuse University, Syracuse, NY 13244, USA; algodine@syr.edu (A.G.); dthill@syr.edu (D.H.); pruthvi.kilaru@dmu.edu (P.K.); srun01@syr.edu (S.R.); blkmush@syr.edu (B.L.K.); 2Department of Environmental Sciences, College of Environmental Sciences and Forestry, State University of New York, Syracuse, NY 13210, USA; btdandar@syr.edu; 3Department of Environmental Biology, College of Environmental Sciences and Forestry, State University of New York, Syracuse, NY 13210, USA; hgreen@esf.edu; 4College of Osteopathic Medicine, Des Moines University, Des Moines, IA 50312, USA; 5Department of Biochemistry and Molecular Biology, SUNY Upstate Medical University, Syracuse, NY 13210, USA; middletf@upstate.edu

**Keywords:** wastewater surveillance, wastewater-based epidemiology, COVID-19, SARS-CoV-2, college dormitories, residence halls, sensitivity analysis, specificity analysis

## Abstract

A residential building’s wastewater presents a potential non-invasive method of surveilling numerous infectious diseases, including SARS-CoV-2. We analyzed wastewater from 16 different residential locations at Syracuse University (Syracuse, NY, USA) during fall semester 2020, testing for SARS-CoV-2 RNA twice weekly and compared the presence of clinical COVID-19 cases to detection of the viral RNA in wastewater. The sensitivity of wastewater surveillance to correctly identify dormitories with a case of COVID-19 ranged from 95% (95% confidence interval [CI] = 76–100%) on the same day as the case was diagnosed to 73% (95% CI = 53–92%), with 7 days lead time of wastewater. The positive predictive value ranged from 20% (95% CI = 13–30%) on the same day as the case was diagnosed to 50% (95% CI = 40–60%) with 7 days lead time. The specificity of wastewater surveillance to correctly identify dormitories without a case of COVID-19 ranged from 60% (95% CI = 52–67%) on the day of the wastewater sample to 67% (95% CI = 58–74%) with 7 days lead time. The negative predictive value ranged from 99% (95% CI = 95–100%) on the day of the wastewater sample to 84% (95% CI = 77–91%) with 7 days lead time. Wastewater surveillance for SARS-CoV-2 at the building level is highly accurate in determining if residents have a COVID-19 infection. Particular benefit is derived from negative wastewater results that can confirm a building is COVID-19 free.

## 1. Introduction

Environmental surveillance of infectious diseases is a complementary aspect to clinical surveillance. Clinical surveillance relies on the identification of individual cases at doctors offices or hospitals to infer community-level information on disease transmission. Environmental surveillance infers community-level information on disease transmission by testing the environment for evidence of the pathogen [1,2]. A community’s wastewater provides a representative sample of that community, and many infectious disease pathogens can be found in wastewater [3]. By surveilling a community’s wastewater, we can gain insight into the infectious diseases that are circulating in that community. Historically, wastewater surveillance has been widely applied to polio eradication efforts [4], as well as antimicrobial resistance [5], and illicit drug use [6].

As the COVID-19 pandemic spread, SARS-CoV-2 (the virus that causes COVID-19) inactive viral fragments were detected in early 2020 at a wastewater treatment plant in the Netherlands [7]. Since this first proof of concept established that inactive viral fragments of SARS-CoV-2 can be measured and quantified in wastewater, wastewater surveillance for SARS-CoV-2 has seen broad adoption around the globe [8]. When analyzing wastewater from municipal treatment plants, wastewater surveillance guides public health policy by providing an unbiased estimate of transmission [9], ensures that transmission is controlled in a community when SARS-CoV-2 RNA is not found [10], and can forecast hospitalizations [11]. Wastewater surveillance can also be conducted upstream from the wastewater treatment plant, wherever wastewater can be sampled including the effluent wastewater from dormitories, prisons, and apartment buildings.

Numerous American universities and colleges incorporated wastewater surveillance into their reopening plans for fall semester 2020 [12]. The most publicized of these was the University of Arizona, where wastewater surveillance detected an outbreak of COVID-19 and allowed early intervention [13]. Testing the residents is perhaps the most logical public health response to detecting SARS-CoV-2 in wastewater. Some universities even locked down dormitories when SARS-CoV-2 was found in wastewater until results from individual testing could be returned [14].

Herein, we examine the efficacy of wastewater surveillance of dormitories to support the COVID-19 response at Syracuse University during the fall of 2020. We determine the sensitivity and the specificity of wastewater surveillance for COVID-19 at the building level, compared to finding cases through individual clinical surveillance.

## 2. Methods

### 2.1. Setting

Syracuse University is situated on an urban campus, directly adjacent to the urban area of the city of Syracuse, New York (population approx. 140,000) and typically enrolls over 20,000 students per year primarily through in-residence instruction. During fall semester 2020 (24 August–24 December 2020), only 15,500 of the 21,322 students enrolled in the university (72%) were considered fully in-person. First-year students are required to live in dormitories on campus, and approximately one-third of undergraduate and graduate students who attend Syracuse University live on campus. On-campus residency at Syracuse University is divided into two main areas, north campus and south campus. North campus consists of large-building dormitories with dining halls, whereas south campus primarily consists of smaller apartment units with their own kitchens. During fall semester 2020, the north campus residence hall population was 4127 and the south campus residency population was 1621, comprising in total 37% of the fully in-person student population.

In response to the COVID-19 pandemic, the university closed its campus during spring break in March of 2020. The university restarted on-campus activities for fall semester 2020, with numerous precautions in place. The fall semester started in mid August, earlier than typical, and ended just before Thanksgiving (24 November 2020). Face masks were required on campus (indoors and outdoors) during fall semester. Students were required to have a negative pre-arrival SARS-CoV-2 test (polymerase chain reaction [PCR] required); and then upon arrival, students were required to test for SARS-CoV-2 weekly for three weeks. Classes were socially distanced, and attendance was limited to either 30 persons or 50% of a room’s maximum capacity, whichever number was lower. Students could attend remotely for almost every class offered, and many classes were completely online. The university also implemented contact tracing including the isolation of cases and the quarantine of contacts, and a wastewater surveillance program. Students residing in dormitories who tested positive for SARS-CoV-2 were removed to the isolation dormitory on south campus, and isolated for 10 days either from the onset of symptoms or their positive test. Students residing in dormitories who were exposed to a SARS-CoV-2 case were removed to the quarantine dormitory off-campus (not included in wastewater surveillance), and quarantined for 14 days.

### 2.2. Wastewater Testing

Wastewater surveillance was a key component of the university’s fall 2020 semester opening plan. Unfortunately, the university does not have its own wastewater treatment plant, nor a single main sewage line that serves the entire campus. However, all 21 dormitory buildings were inspected for the feasibility of routine wastewater sample collection. Of the 18 residence halls owned and operated by Syracuse University on the north campus, wastewater samples were initially collected from 15. Three sampling points had persistent plumbing issues, and so were abandoned, resulting in 12 sampling points on north campus by the end of the semester (Figure 1).

South campus is located approximately two miles south of the main campus and unlike the main campus, which includes both residence halls and academic buildings, south campus is mainly used for student housing, with only a few office buildings (Figure 2). South campus includes 125 student apartments owned by the university that were included in wastewater testing and four privately operated apartment buildings that were not included in the wastewater testing. Since there were over 100 buildings housing students on south campus, building level surveillance was considered impractical and replaced by sampling at four separate points (Figure 2). Two points at the east and west end of south campus, respectively, were chosen and they captured roughly half of south campus excluding the isolation dormitory for COVID-19 cases. A third sampling point outside the isolation dormitory was also included, and an overall south campus sampling point that collected from residential and office buildings, including the COVID-19 isolation dormitory, was the fourth south campus sampling location (see Figure 2 for details).

Automated composite wastewater samplers using peristaltic pumps and marine batteries were used for sampling as detailed elsewhere [15]. Wastewater samplers were operated for twenty-four hours beginning in the morning on Mondays and Wednesdays, and samples were collected in the morning on Tuesdays and Thursdays. Samplers were programmed to deposit a 10 mL sample every 15 min into a 4.55 L jar in an insulated container. Ice was used to keep the collection jar at four degrees Celsius to preserve the integrity of viral nucleic acids. On collection days, samples would be collected and transferred from the 4.55 L jars into eight-ounce bottles, which were then placed on ice and delivered to Quadrant Biosciences (Syracuse, NY, USA) for analysis. Levels of SARS-CoV-2 RNA in wastewater samples were concentrated using ultracentrifugation through a 50% sucrose cushion [16]. Briefly, a 20 mL aliquot from a shaken wastewater sample was transferred to the ultracentrifuge tube. Twelve milliliters of 50% sucrose solution was then carefully pipetted underneath wastewater to form two distinct layers. Samples were balanced using distilled water and ultracentrifuged in a Sorvall WX Ultra series ultracentrifuge equipped with a Sorvall SureSpin 630 (6 × 36 mL) swinging bucket rotor (Thermo Scientific, Waltham, MA, USA) at 150,000× *g* for 45 min. The resulting supernatant was discarded and the pellets were resuspended in 200 μL 1X PBS buffer and transferred to 1.7 mL microcentrifuge tube. Extraction of total nucleic acids was performed with the ZYMO Quick-RNA Fungal/Bacterial Microprep Kit (Zymo, Irvine, CA, USA) eluting in 30 uL elution buffer. Quantification of the RdRp region of SARS-CoV-2 was performed with qPCR on a CFX Real-time PCR Detection System (BioRad, Hercules, CA, USA) using the IP2/IP4 primer sets [17]. crAssphage DNA was quantified using the CPQ_056 assay published previously. Reactions were run in triplicate. The limit of quantification of SARS-CoV-2 RNA using this method is 5 gene copies per mL. Methodological details on this workflow are presented elsewhere [16]. A summary of all wastewater sampling results is collected in Table 1.

### 2.3. Case Data

Throughout fall semester 2020, free COVID-19 diagnosis was offered to students on campus at the Barnes Health Center. In addition, multiple rounds of mass surveillance testing were conducted, including three rounds at the beginning of fall semester and rounds prior to the Halloween (31 October) and Thanksgiving (26 November) holidays. Additionally, students in dormitories were directed to test upon detection of cases in the same dormitory or upon detection of SARS-CoV-2 RNA in wastewater. Any students in on-campus housing that tested positive were moved immediately to the isolation dormitory. We matched de-identified COVID-19 cases to wastewater samples based upon the date of diagnosis and the dormitory building or apartment of the case, irrespective of any potential pre-diagnosis shedding.

### 2.4. Data Analysis

Wastewater results were categorized by how much SARS-CoV-2 RNA was detected in the sample into one of three categories: non-detection, wherein no SARS-CoV-2 RNA was detected; detection but below the level of quantification, wherein SARS-CoV-2 RNA was detected in at least one of the three reactions and/or below the limit of quantification of 5 genomic copies per mL; and quantifiable, wherein the amount of SARS-CoV-2 RNA was above 5 copies per mL. We then created two separate outcomes for dormitory status, the first being detected, wherein a sample from the dormitory was positive for any amount of SARS-CoV-2 RNA in any of three qPCR reactions (non-quantifiable or quantifiable), and the second being quantifiable, wherein a sample had sufficient SARS-CoV-2 RNA that the number of genomic copies could be quantified. For dormitories with two sampling points, we took the highest result from the sampling day. For example, if for a dormitory with two sampling points one sample was of quantifiable level and the other sample was not detected, we classified that dormitory for that sampling date as quantifiable. Dormitory halls were classified into negative, wherein no case was detected, and positive, wherein at least one COVID-19 case was active.

Traditionally, sensitivity, specificity, positive predictive value (PPV), and negative predictive value (NPV) refer to the ability of a test to correctly classify an individual as infected or not. In our study, we use these measures to estimate the ability of wastewater surveillance to classify a residence hall as having a COVID-19 case or not. Sensitivity, specificity, PPV, and NPV are calculated using Equations (1)–(4). As shown in Table 2, a true positive would be a result where clinical surveillance found a COVID-19 case and wastewater surveillance detected or quantified SARS-CoV-2 RNA. A false positive would be a result where clinical surveillance found no COVID-19 case but wastewater surveillance detected or quantified SARS-CoV-2 RNA. A true negative would be a result where clinical surveillance found no COVID-19 case and wastewater surveillance did not detect or quantify SARS-CoV-2 RNA. A false negative would be a result where clinical surveillance found no COVID-19 case but wastewater surveillance did detect or quantify SARS-CoV-2 RNA.
(1)$$\mathrm{Sensitivity}=\frac{\mathrm{True}\;\mathrm{positives}}{\mathrm{True}\;\mathrm{positives}+\mathrm{false}\;\mathrm{negatives}}\times 100$$
(2)$$\mathrm{Specificity}=\frac{\mathrm{True}\;\mathrm{negatives}}{\mathrm{True}\;\mathrm{negatives}+\mathrm{false}\;\mathrm{positives}}\times 100$$
(3)$$\mathrm{PPV}=\frac{\mathrm{True}\;\mathrm{positives}}{\mathrm{True}\;\mathrm{positives}+\mathrm{false}\;\mathrm{positives}}\times 100$$
(4)$$\mathrm{NPV}=\frac{\mathrm{True}\;\mathrm{negatives}}{\mathrm{True}\;\mathrm{negatives}+\mathrm{false}\;\mathrm{negatives}}\times 100$$

We considered lead times between the date of the wastewater sample and the date of the potential COVID-19 case ranging from the same day up to the wastewater sample being taken 7 days prior to the potential date of the COVID-19 case. We calculated the sensitivity, specificity, positive predictive and negative predictive values using the epiR package in R version 4.0.3 [18,19].

## 3. Results

### 3.1. COVID-19 Cases at Syracuse University during Fall Semester 2020

COVID-19 cases at Syracuse University were first identified during pre-arrival and arrival testing (Figure 3). A small surge in cases occurred a few weeks after, which was contained through contact tracing. A second surge in cases occurred following the Labor Day holiday weekend (Monday 7 September 2020), with cases declining until the Halloween holiday (31 October 2020). In response to this Halloween surge, the university suspended all in-person activities on 12 November, and as planned, campus was closed on 21 November prior to the Thanksgiving holiday (Figure 3).

By the end of fall semester 2020, a total of 584 positive COVID-19 cases were identified at Syracuse University with 136 being among dormitory residents (23%). Of these 136 positive cases, 110 resided in north campus residence halls and 26 resided in south campus apartments. There was a high variation in COVID-19 cumulative incidence by dormitory, ranging from 0.9% to 12.1% of residents. In comparison, the cumulative incidence for all Syracuse University students during Fall 2020 semester was 3.8%.

### 3.2. Characteristics of Wastewater Samples Collected

During fall semester 2020, a total of 324 viable wastewater samples were collected. A viable wastewater sample had enough liquid collected to be analyzed; sometimes the samplers would not collect enough liquid and those samples could not be analyzed. Samples came from both north campus (*n* = 255) and south campus (*n* = 69), of which a total of 156 (48%) samples contained detectable levels of SARS-CoV-2 viral RNA material and 100 (31%) contained quantifiable levels of SARS-CoV-2 RNA material (Figure 4).

### 3.3. Sensitivity and Positive Predictive Value of Wastewater Surveillance to Detect COVID-19 Cases for Each Sampling Point

We observed a sensitivity of 95% (95% confidence interval [CI] = 76–100%) and 76% (95% CI = 53–92%) for wastewater surveillance to identify a dormitory with COVID-19 cases diagnosed on the same day as the wastewater sample was collected when considering levels of SARS-CoV-2 RNA in wastewater that were detectable and quantifiable, respectively (Figure 5A). When considering more lead time, wastewater surveillance displayed lower sensitivity. Detectable levels of SARS-CoV-2 RNA in wastewater had a sensitivity of 73% (95% CI = 61–83%) for identification of dormitories with COVID-19 cases up to a week out and quantifiable levels of SARS-CoV-2 RNA in wastewater had a sensitivity of 48% (95% CI = 35–60%) for identification of dormitories with COVID-19 cases up to a week out.

The probability that a positive wastewater test result correctly indicated a COVID-19 case in the dormitory on the same day as testing, or PPV, was 20% (95% CI = 13–30%) for detectable levels of SARS-CoV-2 in wastewater and 30% (95% CI = 18–44%) for quantifiable levels of SARS-CoV-2 in wastewater (Figure 5C). When considering cases up to a week beyond the wastewater test, PPV of correctly identifying a dormitory with a COVID-19 case was 50% (95% CI = 40–60%) for detectable levels of SARS-CoV-2 RNA in wastewater and 59% (95% CI = 45–72%) for quantifiable levels of SARS-CoV-2 RNA in wastewater.

We observed a specificity of 60% (95% CI = 52–67%) and 80% (95% CI = 74–86%) for wastewater surveillance to identify a dormitory without COVID-19 cases diagnosed on the same day as the wastewater sample was collected when considering levels of SARS-CoV-2 RNA in wastewater that were detectable and quantifiable, respectively (Figure 5B). When considering more lead time, wastewater surveillance displayed higher specificity. Detectable levels of SARS-CoV-2 RNA in wastewater had a specificity of 67% (95% CI = 58–74%) for identification of dormitories without COVID-19 cases up to a week out and quantifiable levels of SARS-CoV-2 RNA in wastewater had a specificity of 85% (95% CI = 78–90%) for identification of dormitories without COVID-19 cases up to a week out.

The probability that a negative wastewater test correctly indicated no COVID-19 cases in the dormitory on the same day as testing, or NPV, was 99% (95% CI = 95–100%) for detectable levels of SARS-CoV-2 in wastewater and 97% (93–99%) for quantifiable levels of SARS-CoV-2 in wastewater (Figure 5D). When considering cases up to a week beyond the wastewater test, NPV for correctly identifying a dormitory without COVID-19 cases was 84% (95% CI = 77–91%) for detectable levels of SAR2 RNA in wastewater and 78% (95% CI = 71–84%) for quantifiable levels of SAR2 RNA in wastewater.

## 4. Discussion

We observed high sensitivity, specificity, and NPV as well as a moderate PPV for wastewater surveillance to correctly identify the presence or absence of cases of COVID-19 in dormitories. Whereas sensitivity and NPV decreased with increasing lead time between wastewater tests and COVID-19 cases, specificity and PPV increased with increasing lead time between wastewater test and COVID-19 cases. Taken together, these results suggest that wastewater surveillance at the building level can provide insight into individual infections of the residents in that building with some lead time, indicating early warning of COVID-19 cases. Specificity increased with the number of days prior to the sample of wastewater being tested, with tests seven days prior having 67 percent specificity for detected results and 85 percent specificity for quantifiable levels (see Figure 4). PPV also increased with greater lead time; we found that tests seven days prior had 50 percent PPV for detected levels and 59 percent PPV for quantifiable levels compared to 20 percent and 30 percent PPVs for samples from the same day, respectively. This suggests that wastewater testing may be a good leading indicator of disease with high value for indicating a building has no COVID-19 cases.

For SARS-CoV-2, negative wastewater test results from a residential building are highly predictive of no COVID-19 cases and help in establishing freedom from disease within the residential building [10]. Positive wastewater test results are highly sensitive to identifying cases, but their positive predictive value appears substantially muted. However, the low PPV of wastewater test results could be influenced by a failure of clinical testing to detect the virus when present in any individual in a dormitory. Under this study design, this failure would artificially decrease the number of true positives and result in artificially low PPV for wastewater. We suspect that post-infection shedding is also a significant source of SARS-CoV-2 RNA in wastewater, which would also lower the PPV and complicate the understanding of detected levels of SARS-CoV-2 RNA. Using a higher threshold of the amount of SARS-CoV-2 RNA in wastewater can help mediate this challenge, such as we did with quantifiable levels.

Wastewater surveillance was more sensitive to identifying COVID-19 cases at Syracuse University than Emory University and showed greater NPV (lower PPV) than the University of Notre Dame. At Emory, passive Moore swab sampling and then a skimmed milk laboratory method were poor for detecting individual sporadic dormitory cases (9.5% sensitivity), but generally useful as an indicator for surging campus cases [20]. At Notre Dame, they estimated the sensitivity and the specificity of reverse transcription loop-mediated isothermal amplification (RT-LAMP) to standard PCR laboratory analysis for wastewater surveillance, but did not estimate the sensitivity and the specificity of either method to detect COVID-19 cases [21]. PPV and NPV are highly influenced by disease transmission intensity, and as such are not useful for comparison across settings.

In general, these results corroborate findings from other universities regarding the utility of wastewater surveillance of dormitory buildings to support the response to COVID-19. At Tulane University in New Orleans, the levels of SARS-CoV-2 RNA found in wastewater correlated strongly with clinical incidence of the students [22]. At the University of North Carolina at Charlotte, three-times-weekly wastewater surveillance enabled the identification and isolation of asymptomatic COVID-19 cases [14]. Dormitory-level surveillance can also be used to monitor the emergence and spread of variants of concern, as shown at the University of Nevada, Las Vegas [23]. Additional benefits from wastewater surveillance at universities are not always seen—at Rutgers University and Notre Dame, wastewater surveillance was found to have minimal added benefit to weekly saliva testing of all students [24,25]. Both these campuses required weekly saliva testing of all students, an intervention paradigm that is costly and logistically difficult to replicate. For example, wastewater surveillance of dormitories was estimated to be only 1.7% of the total cost of mass clinical surveillance at Arizona State University [26].

### Limitations

A number of limitations are present in this study. Wastewater samples were only collected twice per week (Tuesdays and Thursdays), leaving a gap in surveillance through the weekends and thus not aligning with potential COVID-19 cases that could be diagnosed any day of the week. Additionally, clinical COVID-19 testing was not systematic throughout the semester. The majority of the semester clinical COVID-19 testing was dependent upon treatment-seeking behavior. In some ways, it would have been better to have systematic and routine clinical COVID-19 testing in order to determine the sensitivity of wastewater surveillance at the building level. However, once a case is identified, that person is removed from the building under surveillance and moved to the isolation dormitory. Isolation of the case would therefore prevent any assessment of the sensitivity of the wastewater surveillance under more natural conditions. Although we used homemade wastewater samplers, we do not expect them to have any impact on the results of this study [15]. Lastly, these results were obtained during fall 2020 semester before the rise of the Alpha, Beta, Delta, and Omicron variants. We hypothesize that different variants of SARS-CoV-2 will not greatly affect the results given what we have seen from wastewater surveillance throughout New York State communities, but further corroboration is needed.

## 5. Conclusions

Wastewater surveillance at the residence hall level has the capacity to guide and inform the public health response to the COVID-19 pandemic. The greatest value was in the non-detection of SARS-CoV-2 RNA in wastewater, with the probability of a negative test predicting zero cases up to a week later exceeding 80%. When SARS-CoV-2 RNA is detected in wastewater from a residential dormitory, it needs to be considered in the context of post-infection fecal shedding. Still, sensitivity to find incident COVID-19 cases exceeding 60% for non-disruptive surveillance is quite good [27]. We expect these results to generalize to other congregate living facilities including high-risk facilities such as nursing homes and prisons [28].

## Figures and Tables

**Figure 1 ijerph-19-04851-f001:**
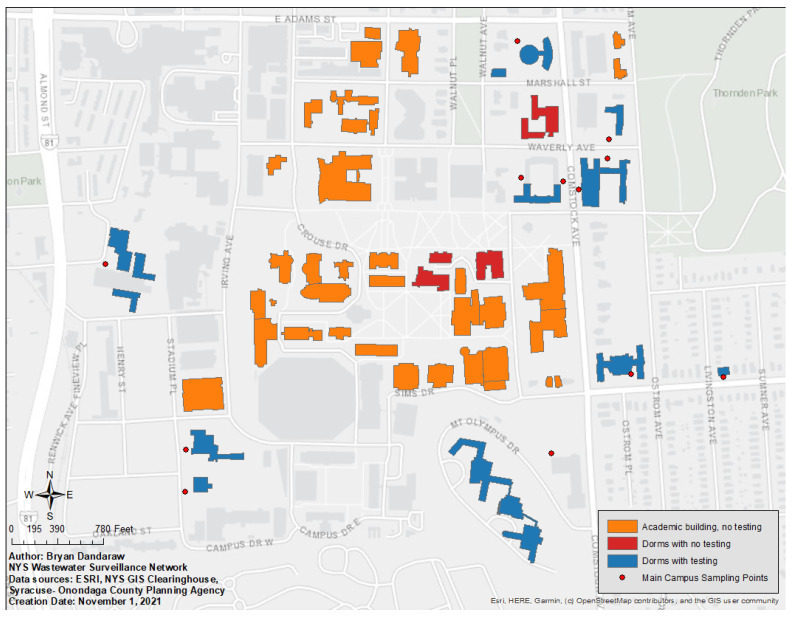
Syracuse University north campus. Buildings are classified as academic or residential (“dorms”) with or without testing fall 2020.

**Figure 2 ijerph-19-04851-f002:**
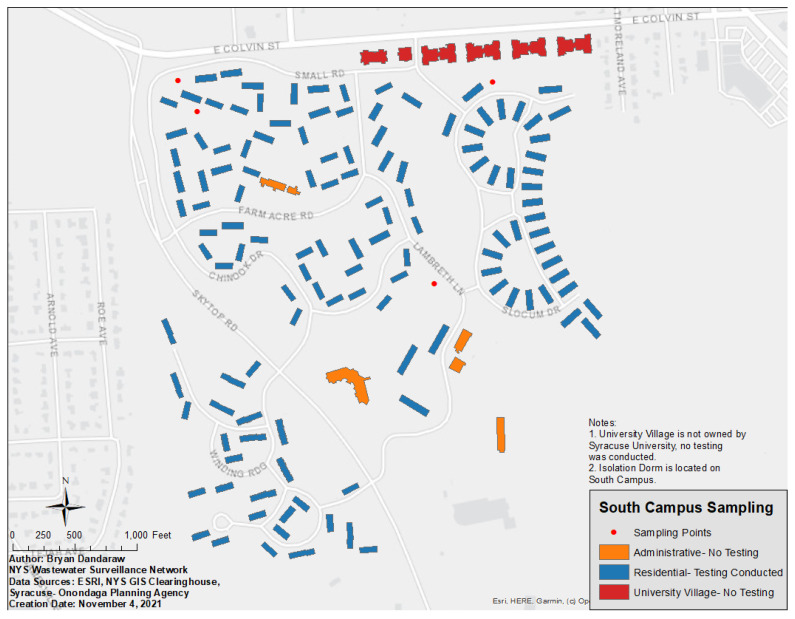
Syracuse University south campus. Sampling points collected wastewater from multiple buildings through shared sewer systems.

**Figure 3 ijerph-19-04851-f003:**
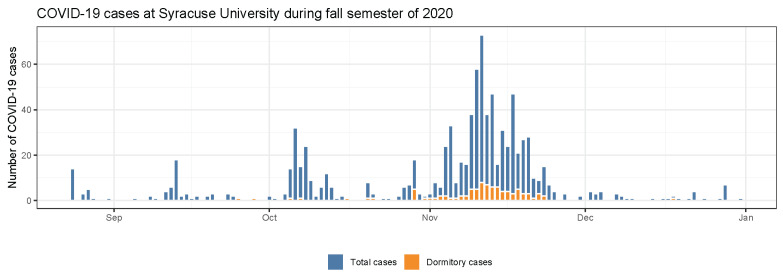
Daily COVID-19 cases during fall semester 2020 at Syracuse University. The first spikes in August represent COVID-19 cases discovered during pre-arrival and arrival testing.

**Figure 4 ijerph-19-04851-f004:**
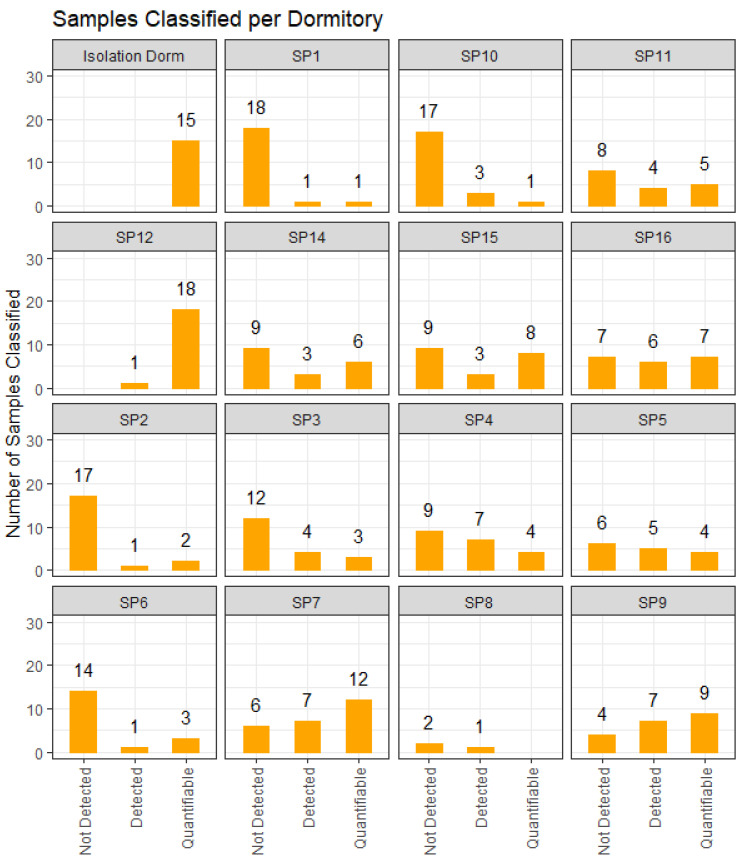
Samples classified by dormitory.

**Figure 5 ijerph-19-04851-f005:**
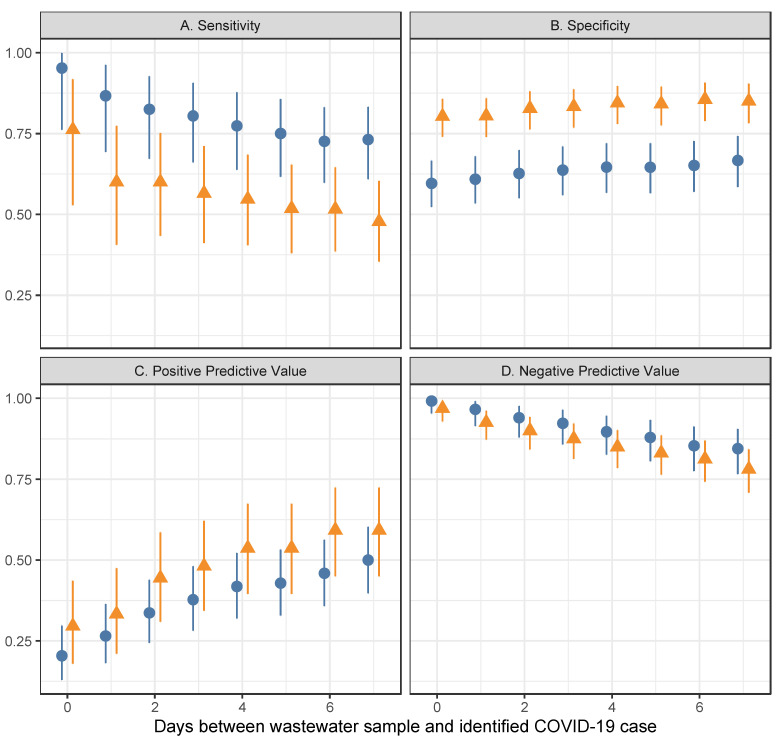
Sensitivity (**A**) to identify a building with a COVID-19 case using wastewater surveillance, specificity (**B**) to identify a building without a COVID-19 case using wastewater surveillance, positive predictive value (**C**) of a positive wastewater surveillance result to indicate a COVID-19 case among residents, and negative predictive value (**D**) of a negative wastewater surveillance result to indicate no COVID-19 cases among residents.

**Table 1 ijerph-19-04851-t001:** Wastewater sample data for each sampling point (SP).

Sampling Point	Number of Observations	Dates Collected	Number Non-Detects	Number of Detects	Min	Mean	Median	Max	Standard Deviation
SP1	20	1 September 2020 to 19 November 2020	18	2	<LOQ	5.45	1	90	19.90
SP2	20	10 September 2020 to 19 November 2020	17	3	<LOQ	4.85	1	75	16.52
SP3	19	4 September 2020 to 17 November 2020	12	7	<LOQ	5.42	1	52	13.32
SP4	20	1 September 2020 to 17 November 2020	9	11	<LOQ	13.85	1	93	30.21
SP5	15	5 September 2020 to 17 November 2020	6	9	<LOQ	11.86	1	86	24.63
SP6	18	1 September 2020 to 17 November 2020	14	4	<LOQ	4.83	1	48	11.36
SP7	25	24 August 2020 to 19 November 2020	6	19	<LOQ	22.56	1	83	29.68
SP8	3	4 September 2020 to 15 September 2020	2	1	<LOQ	1	1	1	0
SP9	20	1 September 2020 to 17 November 2020	4	16	<LOQ	21.65	1	91	31.04
SP10	21	1 September 2020 to 19 November 2020	17	4	<LOQ	1.95	1	21	4.36
SP11	17	7 September 2020 to 12 November 2020	8	9	<LOQ	15.76	1	94	29.51
SP12	19	10 September 2020 to 17 December 2020	0	19	6.97	42.94	38	84	25.91
SP13	15	10 September 2020 to 17 November 2020	0	15	9.82	50.2	49	95	27.89
SP14	18	10 September 2020 to 19 November 2020	9	9	<LOQ	17.33	1	80	26.55
SP15	20	1 September 2020 to 17 November 2020	9	11	<LOQ	20.7	1	89	32.10
SP16	20	4 September 2020 to 19 November 2020	7	13	<LOQ	19.55	1	66	27.27

Notes: The limit of quantification for the method used was 5 copies per mL. All quantitative data are in copies per mL.

**Table 2 ijerph-19-04851-t002:** Classification of wastewater surveillance and clinical detection of COVID-19 cases.

		Clinical Surveillance	
		COVID-19 case	No COVID-19 case
**Wastewater** **surveillance**	SARS-CoV-2 RNA detected or quantified	True positive	False positive
	SARS-CoV-2 RNA not detected or quantified	False negative	True negative

## Data Availability

The data presented in this study are available on request from corresponding author. The data are not publicly available due to privacy restrictions.

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
