# Peer review of "High Sensitivity and Specificity of Dormitory-Level Wastewater Surveillance for COVID-19 during Fall Semester 2020 at Syracuse University, New York"

_ijerph, 2022, doi:10.3390/ijerph19084851_

Round 1

Reviewer 1 Report

The paper titled  High sensitivity and specificity of dormitory-level wastewater 2 surveillance for COVID-19 during fall semester 2020 at Syra-3 cuse University, New Yorkaddressed a good topics, while more corrections are required as following

  1. More details are required for description the analytical methods of covid19 in the wastewater samples
  2. Authors have to explain how the covid 19 can survive in the wastewater
  3. The discussion need more comparison with previous studies to show the differences in different locations

Author Response

Point 1: More details are required for description the analytical methods of covid19 in the wastewater samples.

Response 1: We have added a description of the analytical methods used to detect and quantify SARS-CoV-2 viral fragments in wastewater to the methods section of the paper under a subheading "Wastewater testing". We also reference the appropriate studies that were published previously with greater description of the method used. For this study, we simply purchased laboratory services for wastewater testing rather than conduct the testing in house.

Point 2: Authos have to explain how the Covid 19 can survive in the wastewater

Response 2: Viable SARS-CoV-2 virus has not been found in wastewater to date. Furthermore, we've seen no increased risk of COVID-19 among wastewater treatment plant operators. Inactive fragments of SARS-CoV-2, however, have been found in wastewater and that is what we are presumably measuring in our study. We have specified in the introduction that wastewater surveillance for SARS-CoV-2 detects and measures inactive fragments of SARS-CoV-2 RNA in wastewater.

Point 3: The discussion needs more comparison with previous studies to show the differences in different locations.

Response 3: We have increased the number of similar studies that we cite in the discussion following an updated literature search. Importantly, only one other study reported the sensitivity and specificity of wastewater surveillance to detect cases of COVID-19 in dormitories, despite numerous studies of wastewater surveillance on college campuses. We have incorporated these points into the discussion.

Reviewer 2 Report

The manuscript describes kind of a case study for application of wastewater based epidemiology focusing on the detection of SARS-CoV-2 on a university campus. The study is clearly presented, the manuscript is well written. The study contributes to our knowledge of SARS-CoV-2 detection in sewage and interpretation of findings and will help support the health management of future epidemic or pandemic situations. 

I have only a few minors that should be adressed by the authors.

Abstract: You would monitor pathogens not diseases.

Pls change to SARS-CoV-2 throughout the text (SARS is a disease, SARS-CoV are the viruses that may cause the disease). In general, there is a mix of SARS and Covid. SARS-CoV-2 is the agent. COVID-19 the disease. For example, in line 84 students were tested for SARS-CoV-2 (the agent). It is COVID-19 when you talk about the clinical picture of the infection.

Line 121: Please mention how many liters one gallon is

Line 124: same here: how many ml? You will have international readers.

Line 198 and later: Pls introduce abbreviation in the text.

Line 226: same for NPV

Line 236: there are several reasons to discuss. One for example is individual virus load that may influence detectable CoV RNA in sewage. In addition, different CoV variants lead to different patterns in sewage.

Fig 3: pls re-order panels according to consecutive numbering

Fig 4: Pls add A-D according to text

Do you have any results indicating early detection in wastewater i.e., positive wastewater and no positive individual at the same day but few days later? That would indicate wastewater suitable for early warning system.

Author Response

Point 1:

The manuscript describes kind of a case study for application of wastewater-based epidemiology focusing on the detection of SARS-CoV-2 on a university campus. The study is clearly presented, the manuscript is well written. The study contributes to our knowledge of SARS-CoV-2 detection in sewage and interpretation of findings and will help support the health management of future epidemic or pandemic situations. I have only a few minors that should be addressed by the authors.

Response 1: Thank you for the review of our article.

Point 2: You would monitor pathogens not diseases.

Response 2: Yes we agree with you. We are trying to speak to infectious disease surveillance here, so we have revised to "surveilling" rather than "monitoring".

Point 3: Pls change to SARS-CoV-2 throughout the text (SARS is a disease, SARS-CoV are the viruses that may cause the disease). In general, there is a mix of SARS and Covid. SARS-CoV-2 is the agent. COVID-19 the disease. For example, in line 84 students were tested for SARS-CoV-2 (the agent). It is COVID-19 when you talk about the clinical picture of the infection.

Response 3: Thank you for the clarifications. We have revised throughout. We use SARS-CoV-2 when referencing the virus including testing people and wastewater for the virus. We use COVID-19 to reference cases with the disease and to refer to the pandemic in general.

Point 4: Please mention how many liters one gallon is

Response 4: We have changed over to the metric system. 

Point 5: Line 124: same here: how many ml? You will have international readers.

Response 5: We have changed over to the metric system.

Point 6: Line 198 and later: Pls introduce abbreviation in the text.

Response 6: We have now introduced the abbreviations in the text in the methods section.

Point 7: Line 226: same for NPV

Response 7:  We have now introduced the abbreviations in the text in the methods section.

Point 8: Line 236: there are several reasons to discuss. One for example is individual virus load that may influence detectable CoV RNA in sewage. In addition, different CoV variants lead to different patterns in sewage.

Response 8: These different issues would affect the sensitivity of wastewater surveillance rather than the positive predictive value that we are discussing in these lines. While it is true that individual virus load differs among people, we don't have good studies of fecal viral load throughout the course of infection - all the studies we have currently begin after diagnosis and even hospitalization. It is also true that different variants might have different patterns in wastewater, but we are yet to definitively see that in the scientific literature. We have added a note in the limitations regarding the generalizability of these results beyond fall of 2020.

Point 9: Fig 3: pls re-order panels according to consecutive numbering

Response 9: We have edited Figure 3 to order the panels consecutively.

Point 10: Fig 4: Pls add A-D according to text

Response 10: We have added A-D fore each panel in the figure 

Point 11: Do you have any results indicating early detection in wastewater i.e., positive wastewater and no positive individual at the same day but few days later? That would indicate wastewater suitable for early warning system.

Response 11: Yes, we have detailed these results in Figure 4 and have added to our discussion the significance of these findings. Specificity and positive predictive value both increase with increasing lead time up to seven days.

Reviewer 3 Report

The authors have investigated if wastewater surveillance for SARS-CoV-2 is beneficial for finding COVID-19 cases. The main finding was that a negative wastewater result confirmed that a building is free from COVID-19. The topic is important. The manuscript has many valuable points and is generally well-written. The results seem reliable, but it is nearly impossible for the reviewer to evaluate it, since the data used for the study (especially wastewater results), has not been described in adequate detail. More information is required regarding the wastewater data (number of positives per place, perhaps concentrations, time scale, etc.), which could be shown in tables or figures. Supplementary tables might also be used. The major equations used for data analysis (PPV, NPV) could be added and described in the methods. It is not clear the results of which samples are shown in Figure 3. Below, the authors find more comments for further improvement of the manuscript.

L18 ‘SARS2’ This is not a correct term. Please, correct throughout the manuscript.

Figure 1. It would be good to indicate which dormitories belong to north and which to south. Also, the wastewater sampling points could be indicated.

L111-116 This description remains a bit unclear for the reader. It could be further clarified.

L125 Which were the target genome regions for SARS-CoV-2 in PCR? N1 and/or other? It would be interesting to know some essential details regarding the method and testing.

L145 To my understanding ‘genetic copies’ is not suitable term in this context. ‘Genomic copies’ is often used. Check throughout the manuscript.

L141-160 Check if there are all essential references added.

L165 Labor Day is not celebrated all over the world. Please, add the date.

L180 What does mean a viable wastewater sample?

Figure 3. Which sample data is shown here? What does mean SP? Also, here it could be indicated which belong to north and which to south and how they are linked to wastewater sampling points.

More thorough/extensive discussion is recommended.

Author Response

Point 1: The authors have investigated if wastewater surveillance for SARS-CoV-2 is beneficial for finding COVID-19 cases. The main finding was that a negative wastewater result confirmed that a building is free from COVID-19. The topic is important. The manuscript has many valuable points and is generally well-written.

Response 1: Thank you for the review.

Point 2: The results seem reliable, but it is nearly impossible for the reviewer to evaluate it, since the data used for the study (especially wastewater results), has not been described in adequate detail. More information is required regarding the wastewater data (number of positives per place, perhaps concentrations, time scale, etc.), which could be shown in tables or figures. Supplementary tables might also be used.

Response 2: We appreciate this recommendation from the reviewer. We have added a descriptive summary table stratifying our wastewater data by sample point. We include sample date range, number of non-detects v. positives per location, and summary statistics for quantifiable results.

Point 3: The major equations used for data analysis (PPV, NPV) could be added and described in the methods.

Response 3: We thank the reviewer for this recommendation, and we have added equations and explanations for each. We have also included a table showing the different potential results that we hope clarifies our investigations.

Point 4: It is not clear the results of which samples are shown in Figure 3. Below, the authors find more comments for further improvement of the manuscript.

Response 4: Figure 3 shows all the samples disaggregated by sampling point. We have attempted to clarify this in the manuscript.

Point 5: L18 ‘SARS2’ This is not a correct term. Please, correct throughout the manuscript.

Response 5: We have revised throughout the manuscript.

Point 6: Figure 1. It would be good to indicate which dormitories belong to north and which to south. Also, the wastewater sampling points could be indicated.

Response 6: We have added the sampling points to the map and also added a map of south campus. The original figure 1 was just north campus.

Point 7: L111-116 This description remains a bit unclear for the reader. It could be further clarified.

Response 7: We have updated the description of the south campus sampling locations and why decision to sample groups of buildings was made. We have also added a second map showing the south campus sampling points. This additional context helps report the situation at Syracuse University and why certain sampling points were selected.

Point 8: L125 Which were the target genome regions for SARS-CoV-2 in PCR? N1 and/or other? It would be interesting to know some essential details regarding the method and testing.

Response 9: The laboratory we contracted to test the wastewater uses IP2/IP4 combined to detect SARS-CoV-2. More extensive method description has been provided under the “wastewater testing” subheading of the methods section of the paper including details on quality control.

Point 10: L145 To my understanding ‘genetic copies’ is not suitable term in this context. ‘Genomic copies’ is often used. Check throughout the manuscript.

Response 10: We have switched to using genomic copies in this context.

Point 11: L141-160 Check if there are all essential references added.

Response 11: We have included all the references that we thought were sufficient. If there are additional references the reviewer suggests we will gladly evaluate them.

Point 12: L165 Labor Day is not celebrated all over the world. Please, add the date.

Response 12: We have added the date of the holiday.

Point 13: L180 What does mean a viable wastewater sample?

Response 13: A viable sample was one with enough liquid wastewater collected to be adequately analyzed for SARS-CoV-2 viral concentration. Some samplers did not always produce enough liquid and samples could not be analyzed. We have added this explanation to the text at the appropriate section.

Point 14: Figure 3. Which sample data is shown here? What does mean SP? Also, here it could be indicated which belong to north and which to south and how they are linked to wastewater sampling points.

Response 14: These are all the sample data, from each point. It would not be ethically appropriate for us to identify which results come from which buildings on campus.

Point 15: More thorough/extensive discussion is recommended.

Response 15: We have added a number of points in the discussion, including expansion of the comparison of these results to other universities and colleges. If there are more specific directions on how we should expand the discussion we will attempt to do so.

Round 2

Reviewer 3 Report

The authors have reacted to most comments given regarding the manuscript. It is now easier to follow the study setting and results. Some further comments are given below to further finalize the text.

L142 Change the ‘ul’ to ‘µl’. Check throughout the manuscript.

Table 1. It is good, only the outfit of the table requires finalizing (for instance, big first letters on all titles of the columns)

L215 Check ‘ampler’.

Figure 4. Is sampling point 13 missing? So, the results are based on wastewater samples. This figure is still a bit confusing when it is mentioned that the numbers are classified per dormitory. Would the authors find some way to describe this at general level but keeping it ethical? One point to discuss would be, how well sampling points represent a dormitory, especially in the south campus.

Another issue that could be addressed in discussion or mentioned in the introduction: Is it known if every infected person excretes viruses in feces? Are there some data available in literature?

Author Response

Point 1: L142 Change the ‘ul’ to ‘µl’. Check throughout the manuscript.

Response 1: We’ve made the requested changes.

Point 2: Table 1. It is good, only the outfit of the table requires finalizing (for instance, big first letters on all titles of the columns)

Response 2: We have revised the table to better present it.

Point 3: L215 Check ‘ampler’.

Response 3: Thank you for catching this typo. We have fixed it.

Point 4: Figure 4. Is sampling point 13 missing? So, the results are based on wastewater samples. This figure is still a bit confusing when it is mentioned that the numbers are classified per dormitory. Would the authors find some way to describe this at general level but keeping it ethical? One point to discuss would be, how well sampling points represent a dormitory, especially in the south campus.

Response 4: We have a few dormitories where we had two sampling points. We updated the methods to reflect that we aggregated the samples from these dormitories, taking the higher of the two levels reported.

Point 5: Another issue that could be addressed in discussion or mentioned in the introduction: Is it known if every infected person excretes viruses in feces? Are there some data available in literature?

Response 5: There are studies of fecal shedding, but we are uncomfortable citing them because they have been poorly done. They are done after diagnosis, and oftentimes during hospitalization. They do not reflect the situation at dormitories, and no fecal shedding studies of early infection have been conducted.
